# Discordance among Belief, Practice, and the Literature in Infection Prevention in the NICU

**DOI:** 10.3390/children9040492

**Published:** 2022-04-01

**Authors:** Hossam S. Alslaim, Jonathan Chan, Fozia Saleem-Rasheed, Yousef Ibrahim, Patrick Karabon, Nathan Novotny

**Affiliations:** 1Department of Surgery, Medical College of Georgia at Augusta University, Augusta, GA 30912, USA; hussamasslayem@gmail.com; 2Department of Emergency Medicine, St. John’s Riverside Hospital, Yonkers, NY 10701, USA; jonathancchan30@gmail.com; 3Section of Neonatology, Beaumont Children’s, Royal Oak, MI 48073, USA; fozia.saleemrasheed@beaumont.org; 4Oakland University William Beaumont School of Medicine, Rochester, MI 48309, USA; yibrahim@oakland.edu (Y.I.); pkarabon@oakland.edu (P.K.); 5Section of Pediatric Surgery, Beaumont Children’s, Royal Oak, MI 48073, USA; 6Section of Pediatric Surgery, Monroe Carell Jr. Children’s Hospital at Vanderbilt University, Vanderbilt University School of Medicine, Nashville, TN 37232, USA

**Keywords:** NICU, infection prevention, healthcare policy

## Abstract

This study evaluates practices of infection control in the NICU as compared with the available literature. We aimed to assess providers’ awareness of their institutional policies, how strongly they believed in those policies, the correlation between institution size and policies adopted, years of experience and belief in a policy’s efficacy, and methods employed in the existing literature. An IRB-approved survey was distributed to members of the AAP Neonatal Section. A systematic review of the literature provided the domains of the survey questions. Data was analyzed as appropriate. A total of 364 providers responded. While larger NICUs were more likely to have policies, their providers are less likely to know them. When a policy is in place and it is known, providers believe in the effectiveness of that policy suggesting consensus or, at its worst, groupthink. Ultimately, practice across the US is non-uniform and policies are not always consistent with best available literature. The strength of available literature is adequate enough to provide grade B recommendations in many aspects of infection prevention. A more standardized approach to infection prevention in the NICU would be beneficial and is needed.

## 1. Introduction

Neonatal intensive care unit (NICU) admissions have been on the rise in the last two decades [1]. Between 2007 and 2012, the adjusted rate of NICU admissions has increased by 23% to a rate of 77.9 admissions per 1000 live births.

Despite advances in neonatal medicine, infants requiring neonatal intensive care continue to incur substantial morbidity and mortality [2,3]. Nosocomial or healthcare associated infections (HAI) are one of the modifiable factors that contribute to neonatal morbidity and mortality. They are defined by many authors as conditions caused by infectious agents not presented on admission, typically identified on or after day three of the hospital stay [4]. With the exception of group B streptococcal infections and herpes simplex infections, late onset sepsis in the NICU is thought to be mainly hospital-acquired. Reported rates of HAI range from 6% to 50% [5], with an overall mortality between 20% and 80% depending on the risk factors [6]. HAI also carry increased morbidity and prolonged hospital stay, along with an increase in cost of care [7,8]. The major risk factors for neonatal HAI include prematurity and low birth weight, prolonged parenteral nutrition with delays in initiation of enteral nutrition, gastrointestinal surgery, use of broad-spectrum antibiotics, extended ventilator support and intravascular catheterization [7]. Outbreaks of HAI have also been reported and related to understaffing, overcrowding and contamination of instruments [9].

Because of this burden, multiple interventions are routinely employed to reduce this risk. While scientific evidence is well established in support of the effectiveness of a number of those interventions, no national practice guidelines exist for standard, everyday care and interaction with neonates in the NICU, and compliance rates differ significantly among different facilities [10]. Such variations can be attributed to differing factors such as policy variations at the state level or differing insurance requirements. These variations might negatively impact the quality of care provided at those NICUs and undermine efforts for standardization of care at those facilities. Our aim was threefold: (1) to study the practice of everyday infection prevention policies and practitioners’ beliefs in those policies in NICUs throughout the United States; (2) to review the existing evidence of standard infection practices in the NICU; (3) to provide evidence supporting the value of adding a national standardized policy.

## 2. Materials and Methods

### 2.1. Developing the Survey

An IRB-approved survey was distributed to members of Section of Neonatal Perinatal Medicine of the American Academy of Pediatrics. All subjects gave their informed consent for inclusion before they participated in the study. A systematic review of the literature provided the domains of the survey questions (illustrated in Table 1) based on the strength of recommendation. Given the overwhelming acceptance of the empirical evidence backing hand hygiene, we elected to study and survey the methods used to increase adherence to hand hygiene in NICUs rather than reviewing data in support of hand hygiene. All incomplete surveys were excluded. Data were analyzed using student’s t-test, ANOVA, and chi-square tests, as appropriate.

### 2.2. Data Collection

A literature search was performed on MEDLINE and PUBMED. The search was performed on 18 February 2019 and PRISMA guidelines were followed. Keywords are included in Appendix A. The search yielded 3570 studies. Out of the 75 full-text papers reviewed, 59 studies met our inclusion criteria. Please refer to Figure 1 for the PRISMA flowsheet. Abstracts of those studies were reviewed and studies that meet the goal of our review were selected. Inclusion criteria were as follows:(a)Report on practice patterns and practice changing interventions on hand hygiene compliance within the NICU setting.(b)Association between gloves, gowns, cellphones, accessories, and attire with HAI in the NICU.(c)Type of water utilized for neonatal bathing and infection risk. Sink disinfection practice and infection risk.(d)Siblings care policy and visiting restrictions and its impact on HAI.

## 3. Results

### 3.1. General Survey Results

Of the 3500 surveys that were sent out, 364 were returned, yielding a response rate of 10%. The experience of respondents was 0–10 years (37%), 11–20 years (18%), and 21+ years (45%). The mean number of NICU beds was 47 beds and the most common grouping was 41–80 beds (Table 2). Twenty-six percent of respondents represented NICUs in freestanding children’s hospitals while the rest were part of an adult hospital. Seventy-three percent of respondents were in a university or university-affiliated setting. While larger NICUs, defined as NICUs with greater than 40 beds, are more likely to have policies (71% vs. 45% *p* = 0.0002), their providers were less likely to know them (37% vs. 56%; *p* = 0.04). Across all categories, providers tended to agree more with the effectiveness of each intervention when it is an established policy at their institution (*p* < 0.001).

### 3.2. Hand Hygiene, Gowns, and Gloves

#### 3.2.1. Reviewed Literature

A.Hand Hygiene Compliance:

Hand hygiene continues to be one of the most effective and economical methods of infection control and prevention. The large body of evidence that supports its efficacy is well established. Compliance with hand hygiene practice is a major area of focus for all hospitals. Reported data from across practice sites and specialties ranges from 5% to 89%, representing an average compliance of 40% [10]. In the pediatric and neonatal population, reports are similar [11,12]. Many contributing factors to such a low compliance rate have been reviewed and significant efforts are underway globally to improve that number [13,14,15,16]. Concerns for skin irritation, inefficient distribution of basins, and increasing patient responsibilities in demanding ICU settings have all been cited as potential etiologies for low compliance. Above all, time stands out as a significant hurdle for providers. If appropriately performed, a nurse will require an estimated 60 min per shift for hand hygiene alone [17,18,19].

When reviewing the association between type of hygienic compound and compliance, the use of alcohol-based rub was associated with improved compliance [20]. Improved compliance did translate to decreased rates of nosocomial infection in one Russian study [21]. The efficacy of alcohol vs. soap and water has been studied extensively as well and the overwhelming body of evidence supports the use of alcohol-based products [21,22,23,24]. Additionally, antiseptic handwashing was similar in microbial count when compared with alcohol-based hygiene but providers using the antiseptic handwashing had more skin irritation than when using the alcohol-based hygiene [25].

When looking at demographic variances, studies have shown that female providers are more compliant than their male counterparts and nurses are more compliant than physicians [10,26,27]. Further analysis of variances among groups showed a trend towards better compliance in providers who perceived that their supervisors positively valued hand hygiene [28].

Multiple interventions have been employed and studied to increase hand hygiene either directly by measuring compliance or indirectly by measuring the rates of communicable nosocomial infections (Table 3). The table shows that the efficacy of different awareness methods is well established for improving compliance. Those methods include posters and reminders in the NICU about the importance of hand hygiene, clustering of nursing procedures to reduce patient contacts and to increase hygiene, providing real-time feedback, placing screensavers and computer backgrounds that encourage hand hygiene, placing timers above basins and finally, over-basin videos that illustrate appropriate hand hygiene technique. Unfortunately, the studies have been small and the methods implemented in these studies and the way compliance was assessed varies widely, making it challenging to compare the efficacy of each of the interventions to each other.

NICUs with low rates of acquired infections have employed immediate and individual feedback to health care workers on their hand hygiene performance and collective feedback to the neonatal ICU on acquired infection rates, while NICUs with high rates of acquired infections only communicated when there was a problem with the acquired infection rate and did not address improper individual hand-washing performance [42].

Unfortunately, studies that report long-term compliance after the individual implementation of the above-described techniques showed the improvement rate often returned to baseline [43,44]. This return to baseline was as short as 3 months after the intervention in some reports [44]. Instead, successful interventions incorporated a combination of strategies including feedback, visual reminders, and placement of hygiene resources including sinks and hand sanitizer dispensers [43]. These findings informed our survey to not only enquire about compliance protocols in NICU but also address long-term sustainability of such compliance.

B.Gloves and Gowns:

Using clean, non-sterile gloves in addition to hand hygiene has been shown to decrease neonatal infections in three available studies [45,46,47]. In a pre- and post-intervention study that included 200 preterm infants, Janota et al. found that neonatal late onset sepsis incidence decreased from 2.99/1000 hospital days and 54.1/1000 admissions with only hand hygiene to zero cases of late onset sepsis when non-sterile gloves were used in addition to hand hygiene practices [45]. Similarly, in a randomized trial enrolling 120 infants, Kaufman et al. found that the use of non-sterile gloves after hand hygiene resulted in a decrease in both Gram-positive and central-line-associated bloodstream infections in preterm neonates [46]. During RSV season, a retrospective cohort study found that non-sterile gloving resulted in significantly lower rates of bacteremia and central-line-associated bloodstream infections in the pediatric and neonatal ICU setting [47].

To our knowledge, no study has been performed on the influence of gloves limiting the providers ability to interact with the surrounding environment, specifically cell phones, between hygiene application and patient interaction. This lack of interaction with fomites may account for the reported decrease in rates of infection. Additionally, all three studies used gloves after hand hygiene versus simply donning gloves alone. Any implementation of gloved examination must include hand hygiene prior to donning gloves.

While gloves seem to be effective in reducing the risk of neonatal infections, research on gowning in the NICU over the past several decades has not been able to demonstrate a benefit [48,49,50,51,52]. A recent Cochrane review and a Cochrane brief affirmed the same finding [53,54].

#### 3.2.2. Survey Results

There are two goals for any intervention that is directed toward hand hygiene: (1) increasing compliance and (2) attaining sustainability. Table 4 summarizes our survey results in this domain. As was found in the reported literature, many NICUs have implemented similar policies to the ones we have mentioned above; however, the support for those policies varies.

In our survey, 46% of providers reported that non-sterile gloves are standard practice in their neonatal ICUs and less than 5% reported the mandatory use of gowns during general patient care. The level of support and belief in those interventions mirrors their utilization with 52% of the providers supporting non-sterile glove use and only 7% for gowns. Providers were asked to rate their opinion of each intervention on the Likert scale of 1–5 with 1 being strongly disagree and 5 being strongly agree. Results are summarized in Table 5 and showed that providers with more years of experience had lower opinion scores on non-sterile gowns (*p* < 0.01). Additionally, as was the case with hand hygiene practice, if the intervention is an institutional policy, providers are more likely to believe in the efficacy.

### 3.3. Clothing and Attire: White Coats, Ties and Sleeves

#### 3.3.1. Reviewed Literature

Health care providers have expressed concern that white coats may serve as vectors for infection transmission [55] with studies showing Staphylococcus aureus being the most common microbe isolated on provider white coats [56,57].

A limited number of studies exist on the role of health care providers’ ties and sleeves in hospital acquired infections, and none are specific to the context of the NICU. Weber et al. found that while unsecured ties were associated with an increase in transmission of bacteria, there was no relationship between the length of sleeves and rate of transmission [58]. Following the UK’s Bare Below the Elbows policy, which restricted long sleeves, watches, jewelry, and ties for clinical staff, Willis-Owen et al. found that the dress code was not correlated with the quantity of microbes nor the presence of drug-resistant organisms on the hands of health care providers [59]. Farrington et al. also found that the policy was not linked to any improvement in physician handwashing [60].

A recent systematic review including adult patients from 2016 evaluating 72 studies showed large variation in the rates of contamination of personal clothing and white coats in the hospital setting. The rates of contamination in general were between 0 and 32% for MRSA and Gram-negative rods. Of those studies, four studies evaluated for possible connection between healthcare personnel contaminants and clinical isolates with no clear link identified [61].

#### 3.3.2. Survey Results

Our survey reviewed policies, opinions and general trends in relation to attire and Table 6 summarizes our findings.

Providers with less experience (0–5 years) were more likely to believe in those policies than providers with +20 years of experience (*p* < 0.001 for both).

### 3.4. Other Fomites and Accessories

#### 3.4.1. Reviewed Literature

A.Stethoscopes and Cell Phones

The most frequent isolates on stethoscopes are Staphylococcal species including methicillin-resistant Staphylococcus aureus as well as Gram-negative organisms [62].

Stethoscopes rates of colonization are as high as 85% with a reduction to 30% when disinfected with alcohol wipes. For cellphones, ample evidence shows high rates of colonization of up to 85% and correlation between pathogens of hands of parents and providers and the type of pathogens colonizing their cellphones [63,64,65]. Unfortunately, no studies in the neonatal setting attempted to establish a link between stethoscopes as a vector and health associated infections. A recent systematic review in the adult ICUs, however, has shown a correlation [66].

B.Rings, Watches, and Artificial Fingernails

Accessories have also been found to harbor bacteria. Rings have been shown to carry S. aureus along with Klebsiella and fungi [67]. Gram-negative pathogens, S. aureus, and Candida species are associated with artificial fingernails. A study showed a positive correlation between length of nail and the likelihood to isolate a pathogen [68]. Between 40 and 60% of healthcare workers believe that rings and fingernails contribute to infections [68]. In addition, rings and wrist watches have been found to interfere with hand hygiene [69].

#### 3.4.2. Survey Results

In our survey, approximately 45% of respondents reported having policies that required stethoscope disinfection and regulated the use of mobile devices in the ICU. Those regulations varied from complete prohibition of mobile phone use in the ICU (30% of those with policies) to less restrictive measures like pre-entry disinfection (40%), hand disinfection after each use, and placement of mobile devices in plastic bags (40%). Some institutions had overlapping policies and there was a large variation among policies directed towards staff and family members. Seventy percent of respondents had a no jewelry policy in the ICU.

### 3.5. Environment and Facilities

#### 3.5.1. Reviewed Literature

We identified only one study that evaluated the difference in infection rates between private rooms and an open layout in the NICU. They found no difference in rates of methicillin-resistant Staphylococcus aureus colonization, late-onset sepsis, or mortality [70].

Some NICUs have used sterile water and self-disinfecting sinks as a method to decrease Pseudomonas aeruginosa colonization and infection. A single-center study in a NICU experiencing colonization of *P. aeruginosa* revealed that changing from tap water to sterile water for bathing showed a reduction in the incidence of neonatal bacterial infection [71]. A study comparing the introduction of self-disinfecting sinks vs. replacement of colonized sinks showed superiority of self-disinfecting sinks in reducing bacterial bioburden and but without a direct link to reduction in health associated infection [72]. Cost drivers that resulted in an increased ratio of patient beds per hand washing sink also resulted in increased risk-adjusted odds of hospital acquired bacteremia. Finally, a single study found that nosocomial bacteremia is reduced in units with the presence of more washbasins [73].

#### 3.5.2. Survey Results

Our survey showed about 50% of providers were unfamiliar with the policies in their NICU for water sterilization. Of those who were familiar with the policies for environment and facilities, 10% reported strict use of sterile water for bathing and 4% reported the use of self-disinfecting sinks, with the majority being neutral about their efficacy.

### 3.6. Siblings and Family Visitation

#### 3.6.1. Reviewed Literature

While almost every NICU offers full visiting privileges for parents of infant patients, a wide range of visiting policies exist for siblings and other family members [74]. The basis for these discrepancies on visiting policies include facilitating social support of parents of neonates while limiting exposure to infected visitors, especially younger siblings [74]. This concern is not unwarranted. A comparison of hand cultures of nurses and “homemakers” found that “homemakers” had increased numbers and virulence of bacterial colonization on their hands compared to nurses [75]. A case report also described parents as potential vectors for infection transmission among triplet siblings in a neonatal ICU [76]. Peluso et al. found that restricting sibling visitors during respiratory syncytial virus (RSV) season was linked to a decrease in the number of RSV positive and RSV symptomatic neonates in the NICU [77]. Another study failed to show a direct association between siblings visiting and hospital-acquired viral infections [78].

#### 3.6.2. Survey Results

Eighty percent of the surveyed providers reported restrictions for young relatives’ access during specific seasons. Seventy-three percent reported that parents are required to perform hand hygiene between siblings’ interactions and sharing toys between siblings is reported to be prohibited by 55% of responders. The majority of our respondents believe in the efficacy of those measures.

## 4. Discussion

The subject of infection control in NICUs has been the focus of many studies over decades. Those studies have had different outcome parameters, different methodology and have been carried out in different institutional settings. Although the response rate was limited to 10% of the applicable population and thus subject to response bias, our survey confirmed non-uniform practice and found there is disagreement on the effectiveness of each practice among practitioners with different years of experience, hospital setting, and institutional policies. Those two factors limit the ability to draw generalizable conclusions from the reviewed evidence that would be applicable to each and every NICU. Rather, our goal is to provide a framework of organized data that should be utilized to guide policy development at NICU-containing institutions. We outline our recommendations below with the strength of recommendation and level of evidence supporting it.

Grading and recommendations were based on the GRADE approach in the Cochrane Handbook for Systematic Reviews of Interventions [79] and took into account the risks of bias, imprecision, inconsistency, indirectness, confounding bias, and publication bias. Based on this criterion, the grading scale is as follows: A.High certainty of the evidence: The intervention leads to a large reduction/increase in outcome.B.Moderate certainty of the evidence: The intervention leads to a moderate reduction/increase in outcome.C.Low certainty of the evidence: The intervention leads to a small reduction/increase in outcome.

Levels of Evidence were based on the Levels of Evidence for Prognostic Studies [80] and are as following levels:1.High quality prospective cohort study with adequate power or systematic review of these studies.2.Lesser quality prospective cohort, retrospective cohort study, or systematic review of these studies.3.Case–control study or systematic review of these studies.4.Case series.5.Expert opinion; case report; clinical examples.

### 4.1. Physical Barriers: Hand Hygiene, Gowns, and Gloves

Several studies support a variety of methods to increase compliance and achieve sustainability with hand hygiene practice. Since there is not a single ‘silver bullet’, we recommend that each institution tailors its efforts and policies to its cultural trends and providers’ perception. Our survey did show a statistically significant association between institutional policies and providers’ belief in their efficacy. Specifically, for all above interventions, providers are more likely to believe in the efficacy of a policy if it is instituted at their health care system. That argues that providers are open and receptive to institutional directions and as such would be inclined to adhere to them. It is also possible that the providers surveyed were instrumental in the development of each institution’s policies and, therefore, they believe in their efficacy and, therefore, instituted those policies. Conforming those policies to fit into the dynamics of each institution is also paramount. In an academic facility with high provider turnover (rotating residents and other learners), providers and learners might be better served with videos above sinks and individual on point feedback instead of email reminders and grand round presentations.

The only way to achieve sustainability is by periodic interventions and ongoing reassessments. Without those, any intervention aimed at increasing compliance has been shown to lack a long-term effect.

We make a grade B recommendation based on level 2 evidence for:−The use of alcohol-based hand hygiene as opposed to soap-based hygiene to improve compliance.−Structured periodic feedback systems to sustain long term hand hygiene compliance.−A multimodal approach to spreading awareness and improving compliance for hand hygiene; we recommend incorporating visual reminders and individual feedback.−The use of non-sterile gloves in addition to hand hygiene during the care of preterm infants and during the RSV season for all infants.

Based on level 1 evidence that failed to show improved outcomes with use of non-sterile gowns, we make a grade A recommendation that there is no benefit for their routine use.

### 4.2. Attire

While the evidence to support attire policies for infection control is lacking, most institutions employ policies related to attire and a large proportion of providers believe in their efficacy. Multiple studies have shown white coats to be colonized with pathogens classically associated with HAI, but the ones investigating an association between attire and HAI failed to establish a link. The literature has also shown that ‘bare elbows’ did not decrease the number of pathogens on providers’ hands and did not improve the practice of hand washing [58,59,60]. It might be more appropriate that efforts and resources deployed to raise awareness and reinforce these policies be utilized for improving compliance with hand hygiene and other proven methods of infection control.−We make a grade C recommendation based on level 3 evidence for prohibition of unsecured ties in the NICU to decrease infection transmission.−The current level 2 evidence does not support prohibition of white coats or a bare elbow policy to decrease infection transmission in the NICUs. We recommend resource utilization to reinforce other evidence proven interventions.

### 4.3. Fomites and Accessories

While the evidence supports the potential danger of fomites as vectors for transmission, causality in the NICU setting has not been established. This is reflected in the wide variation of polices, or lack thereof, across different neonatal ICU settings. Given established causality between stethoscopes and infection transmission in the adult based practice, it is reasonable to deploy policies that restrict use of cellphones in the ICU and regulate stethoscope disinfection. A trend that is being seen in both the adult and neonatal setting is the use of disposable stethoscopes to limit infection transmission. It is not clear, however, whether they are beneficial and to what extent lower quality stethoscopes might jeopardize the sensitivity of auscultation.

The published evidence on accessories is observational and weak. No study reviews outcomes and the endpoints and none is powered to establish causality—particularly in the NICU setting. It does however suggest that accessories can negatively impact proper hand hygiene and that cellphones are highly colonized with bacteria.

We make a grade C recommendation based on level 3 evidence for:−Disinfecting stethoscopes between patient encounters.−Instituting policies that regulate the use of cellphones in the NICU. Not enough data exists to recommend prohibiting or restricting their use. However, hygiene is recommended after each use.

We make a grade B recommendation for removal of accessories upon entry to the NICU based on level 3 evidence and their established interference with proper hand hygiene.

### 4.4. Environment and Facilities

Not enough evidence exists to recommend a specific ratio of beds to hand-sanitizing stations (alcohol stations or hand-washing sinks) for infection prevention, but the evidence shows a trend towards higher rates of HAI with higher ratios. It appears that the majority of providers are either unaware of their hospital policies in regard to water sterilization or indifferent about them. This is consistent with a lack of strong evidence supporting its use.−We make a grade B recommendation based on level 2 evidence in favor of using sterile water for bathing and self-disinfecting sinks in neonatal ICU settings with high rates of tap water pseudomonal colonization.

### 4.5. Siblings and Family Visitation:

−We make a grade C recommendation based on level 3 evidence for restricting siblings’ access to NICU during RSV seasons.

## 5. Conclusions

Practice and belief of routine infection prevention in the NICU in the United States is non-uniform and non-conforming to the existing evidence for nosocomial infection prevention. While the literature does not have consistent measures or outcomes, there is enough literature to inform practitioners in development of guidelines. A more standardized approach to infection prevention in the NICU would be beneficial and is needed. We acknowledge the differences in implementation may vary based on state policies, third-party providers, and healthcare structures. This standardized approach will allow for all institutions no matter the size, location, or academic affiliation to adhere to the best practices for all NICU patients. This will effectively promote physician recognition of best care practices and thus limit provider-related infection transmission. We hope that this paper will serve as the first step to creating awareness of the variability of the institutional policies, the lack of awareness to some policies, and the potential dangers of not adhering to such policies. Ultimately, we recommend implementing cost effective national guidelines that will effectively reduce infection risk by increasing awareness and adherence to best policies in all NICUs.

## Figures and Tables

**Figure 1 children-09-00492-f001:**
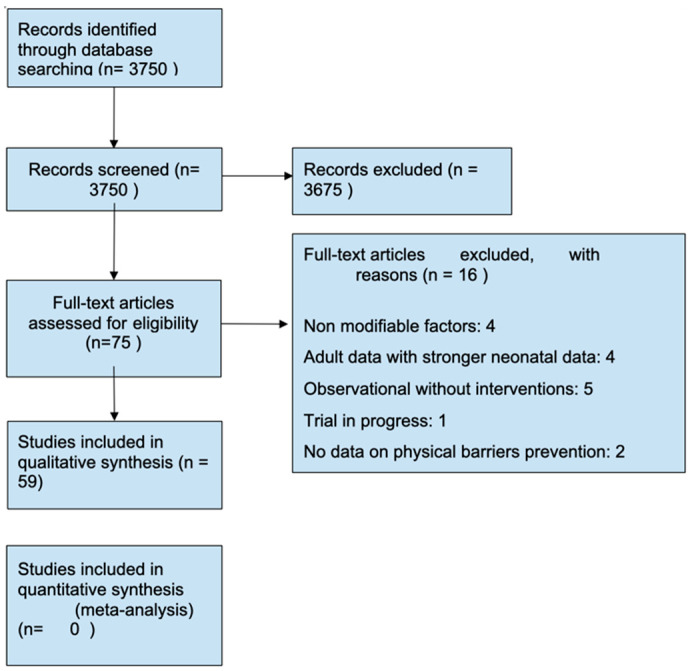
Prisma Flowsheet.

**Table 1 children-09-00492-t001:** The Domains of the Survey Questions Sent to Institutions.

Domain	Description
Demographic data	NICU size, presence of trainees, academic affiliation, years of experience
Physical barriers	Hand hygiene, gloving and gowns
Attire	White coat, neck ties, jewelry, bare elbows
Mobile phones	Regulation of use and practice policy
Environment	Visitors, parents’ hand hygiene, toy sharing
Facility	Sterile water use, self-disinfecting sinks
Hand hygiene efforts	Methods to promote compliance and ensure long-term adoption of hand hygiene

**Table 2 children-09-00492-t002:** Survey Results, Stratified by NICU Bed Size.

Survey Question *	0–20 (*n* = 57)	21–40 (*n* = 119)	41–80 (*n* = 143)	81+ (*n* = 45)	*p*-Value
Hand hygiene policy	57 (100.00%)	118 (99.16%)	143 (100.00%)	45 (100.00%)	0.5591
Non-sterile gloves policy	19 (33.33%)	57 (47.90%)	76 (53.15%)	18 (40.00%)	0.0613
Non-sterile gowns policy	1 (1.75%)	10 (8.40%)	3 (2.10%)	0 (0.00%)	0.0159
Patient-dedicated stethoscope	1 (1.75%)	1 (0.84%)	1 (0.70%)	0 (0.00%)	0.8000
Sterile gloves for <28 weeks GA	0 (0.00%)	1 (0.84%)	0 (0.00%)	0 (0.00%)	0.5591
Bare below elbows	0 (0.00%)	0 (0.00%)	1 (0.70%)	0 (0.00%)	0.6708
No white coat policy	29 (50.88%)	74 (62.18%)	94 (65.73%)	24 (53.33%)	0.1748
No unsecured neck ties policy	21 (36.84%)	44 (36.97%)	53 (37.06%)	11 (24.44%)	0.4379
No jewelry policy	40 (70.18%)	83 (69.75%)	102 (71.33%)	33 (73.33%)	0.9721
Bare bones (short sleeves) policy	37 (64.91%)	82 (68.91%)	101 (70.63%)	29 (64.44%)	0.8038
Pre-Examination stethoscope disinfection policy	30 (52.63%)	55 (46.22%)	61 (42.66%)	19 (42.22%)	0.6037
None of the above 5 policies	8 (14.04%)	10 (8.40%)	9 (6.29%)	5 (11.11%)	0.3351
Young relatives’ access restricted policy	52 (91.23%)	96 (80.67%)	117 (81.82%)	35 (77.78%)	0.2607
Parents perform hand hygiene policy	42 (73.68%)	91 (76.47%)	108 (75.52%)	28 (62.22%)	0.2855
Sharing of toys/utensils is prohibited policy	36 (63.16%)	67 (56.30%)	77 (53.85%)	22 (48.89%)	0.5051
No environmental policies in place	3 (5.26%)	7 (5.88%)	9 (6.29%)	1 (2.22%)	0.7664
Sterile water used for bathing policy	1 (1.75%)	8 (6.72%)	20 (13.99%)	7 (15.56%)	0.0196
Self-disinfecting sink drain policy	1 (1.75%)	6 (5.04%)	4 (2.80%)	2 (4.44%)	0.6499
Do not know facility policies	26 (45.61%)	47 (39.50%)	79 (55.24%)	26 (57.78%)	0.0424
No facility policies (prior 3 rows)	30 (52.63%)	60 (50.42%)	43 (30.07%)	11 (24.44%)	0.0002
Increasing accessibility to alcohol based hand rubs	55 (96.49%)	118 (99.16%)	140 (97.90%)	43 (95.56%)	0.4682
Installing/increasing posters/reminders around NICU	7 (64.91%)	97 (81.51%)	112 (78.32%)	36 (80.00%)	0.0901
Reducing alcohol application time < 30 s	5 (8.77%)	21 (17.65%)	19 (13.29%)	7 (15.56%)	0.4421
Clustering of nursing procedures	46 (80.70%)	85 (71.43%)	93 (65.03%)	29 (64.44%)	0.1404
No compliance policies (prior 4 rows)	1 (1.75%)	1 (0.84%)	1 (0.70%)	1 (2.22%)	0.7937
Periodic performance feedback to promote adoption	26 (45.61%)	75 (63.03%)	84 (58.74%)	28 (62.22%)	0.1602
Periodic audit reports to promote adoption	15 (26.32%)	70 (58.82%)	84 (58.74%)	30 (66.67%)	< 0.0001
Periodic hand hygiene course to promote adoption	16 (28.07%)	38 (31.93%)	36 (25.17%)	16 (35.56%)	0.4811
Courses during grand rounds to promote adoption	9 (15.79%)	22 (18.49%)	22 (15.38%)	9 (20.00%)	0.8505
None of the above to promote adoption (prior 4 rows)	17 (29.82%)	20 (16.81%)	26 (18.18%)	5 (11.11%)	0.0837
Other methods to promote adoption	2 (3.51%)	4 (3.36%)	6 (4.20%)	1 (2.22%)	0.9370
Surgical scrubbing for all is best policy	23 (40.35%)	42 (35.29%)	41 (28.67%)	10 (22.22%)	0.1623
Alcohol based hand rubs is best policy	3 (5.26%)	18 (15.13%)	41 (28.67%)	10 (22.22%)	0.0009
Soap based hand washing is best policy	26 (45.61%)	55 (46.22%)	61 (42.66%)	22 (48.89%)	0.8799
Other things are best policy	6 (10.53%)	9 (7.56%)	9 (6.29%)	5 (11.11%)	0.6395

* all data is reported as n (%).

**Table 3 children-09-00492-t003:** Summary of Articles Studied to Increase Hand Hygiene Either Directly by Measuring Compliance or Indirectly by Measuring the Rates of Communicable Nosocomial Infections.

Country	Intervention	Measured Outcome	Results
US (2002) [29]	Posters, Feedback	Compliance	47% to 85%
China (2004) [30]	Posters, Feedback	Infection rate	17 to 9 per 100 admission
Thailand (2005) [31]	Multilevel	Compliance	35% to 50%
Switzerland (2007) [32]	Posters, Feedback	Compliance	42% to 55%
Netherlands (2011) [33]	Multimodal	Compliance	23% to 50%
Netherlands (2012) [34]	Screensavers	Compliance	63% to 71%
Canada (2013) [35]	Multimodal	Compliance	50 to76%
LMICs (2013) [36]	Multimodal	Compliance	48% to 71%
USA (2013) [37]	Failure mode effectiveness	Compliance	50% to 84%
India (2015) [38]	Posters, Feedback	Sepsis rate	96 to 47 per 1000 patient days
Iran (2015) [39]	Multimodal	Compliance	30% to 70%
Nepal (2017) [40]	Over basin video	Compliance	9% to 68%
US (2018) [41]	Over basin video	Compliance	42% to 72%
Mexico (2019) [26]	Multimodal	Compliance	45% to 79%

**Table 4 children-09-00492-t004:** Survey Results for Hand Hygiene Compliance.

**Compliance Interventions (*n* = 364)**		
**Intervention**	**Institution Utilization**	**Support**
Increasing basin accessibility	97%	98.5%
Posters and reminders	77%	86.1%
Alcohol application < 30 s	14%	40%
Clustering of nursing procedures	69%	88%
**Sustainability interventions (*n* = 364)**		
Performance feedback	58%	84%
Emailed audit reports	54%	78%
Mandatory courses	29%	49%
Grand rounds presentations	17%	39%
None of the above	18%	n/a

**Table 5 children-09-00492-t005:** Likert Opinion Scores, Stratified by Years of Experience.

Survey Question *	0–5 (*n* = 98)	6–10 (*n* = 35)	11–20 (*n* = 67)	20+ (*n* = 164)	*p*-Value
Opinion on effect of hand hygiene	5.00 (0.00)	5.00 (0.00)	4.96 (0.27)	4.99 (0.11)	0.1940
Opinion on non-sterile gloves	3.73 (1.04)	3.69 (0.96)	3.52 (1.12)	3.38 (1.02)	0.0522
Opinion on non-sterile gowns	2.67 (0.88)	2.66 (0.59)	2.49 (0.89)	2.16 (0.86)	<0.0001
Opinion on no white coat in NICU	4.47 (0.79)	4.40 (0.69)	4.33 (0.84)	3.92 (1.09)	<0.0001
Opinion on no unsecured neck ties	4.14 (0.95)	4.00 (0.87)	4.00 (0.85)	3.71 (0.99)	0.0027
Opinion on no jewelry in NICU	4.03 (0.99)	3.94 (1.00)	4.42 (0.82)	4.29 (0.90)	0.0105
Opinion of bare elbows in NICU	4.27 (0.86)	4.29 (0.83)	4.19 (0.94)	4.30 (0.86)	0.8532
Opinion of pre-examination stethoscope disinfection	4.27 (0.86)	4.11 (0.90)	4.15 (0.93)	4.26 (0.87)	0.6911
Opinion of limiting Young relatives’ access to NICU	4.43 (0.81)	4.34 (0.91)	4.22 (1.03)	4.26 (0.98)	0.4597
Opinion of parents performing hand hygiene	4.60 (0.67)	4.34 (0.80)	4.64 (0.64)	4.52 (0.69)	0.1614
Opinion of not allowing shared toys or utensils	4.31 (0.88)	3.97 (0.89)	4.39 (0.85)	4.26 (0.83)	0.1253
Opinion of sterile water being used for baths	2.97 (0.95)	3.09 (0.85)	2.82 (0.85)	2.91 (0.79)	0.4692
Opinion of self-disinfecting sink drains in NICU	3.34 (0.73)	3.26 (0.66)	3.21 (0.57)	3.23 (0.67)	0.5783
Opinion of increasing accessibility to alcohol based hand rubs	4.92 (0.28)	4.86 (0.43)	4.91 (0.29)	4.78 (0.52)	0.0378
Opinion of installing/increasing posters/reminders around NICU	4.26 (0.80)	4.37 (0.77)	4.54 (0.64)	4.24 (0.85)	0.0588
Opinion of reducing alcohol application time below 30 s	3.51 (0.82)	3.60 (0.88)	3.64 (0.85)	3.38 (0.81)	0.1161
Opinion of clustering of nursing procedures to reduce patient contacts	4.40 (0.74)	4.20 (0.76)	4.30 (0.84)	4.17 (0.87)	0.1705
Opinion of periodic performance feedback w/face to face interaction	4.16 (0.74)	4.31 (0.68)	4.45 (0.68)	4.26 (0.83)	0.1296
Opinion of distributing periodic audit reports	4.05 (0.83)	4.17 (0.75)	4.22 (0.78)	4.09 (0.80)	0.5375
Opinion of effectiveness of periodic mandatory hand hygiene courses	3.30 (1.07)	3.51 (0.98)	3.66 (1.05)	3.60 (0.97)	0.0696
Opinion of hand hygiene presentations during grand rounds	3.27 (1.05)	3.31 (1.11)	3.45 (0.97)	3.28 (0.99)	0.6674

* All data is reported as Median (SD).

**Table 6 children-09-00492-t006:** Policies, Opinions and General Trends in Relation to Attire.

Policy	Institutional Utilization (*n* = 364)	Support (*n* = 364)
No white coat	60%	79%
No unsecured ties	35%	64%
No jewelry	70%	80%
Bare elbows	68%	80%

## Data Availability

Data is available by contacting the publishing authors.

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
