# Peer review of "Discordance among Belief, Practice, and the Literature in Infection Prevention in the NICU"

_children, 2022, doi:10.3390/children9040492_

Round 1
Reviewer 1 Report
It is a very interesting and quite useful subject because Infection prevention is one of the most important in the NICU. I believe this study will serve as reducing infection in NICU.
There are only minor comments that I would like to discuss with the authors.
1. Page 1 : HAI and HAIs are mixed. Are HAIs Plural of HAI? Author already described HAI as an abbreviation of Healthcare associated infections.
2. Page 2 : Does HPIs on the 2nd line correct?
3. Table 5 : Is the content correct about ‘Opinion of Young relatives’ access to NICU’ and ‘ Opinion of sharing toys or utensils’ in Table 5? I believe they need to be more clear.
4. Page 11: I believe a reference about Level of evidence (LOE) is needed to help readers understand.
Thank you for the possibility to read this manuscript.
Reviewer 2 Report
The authors present a survey evaluating practice and belief of infection prevention practices and policies among NICUs in the United States. Additionally, the authors performed a systematic review of the literature on this topic. Ultimately, the authors suggest efforts focused on national protocol/policy efforts are warranted due to differences among each institution. Overall, it is a well-written manuscript, but it could be improved with a few minor revisions:
1) I am unclear about the denominator of who responded to the survey. As the survey was sent to members of the Section of Neonatal Perinatal Medicine of the AAP, the authors received 364 responses. However, what is the response rate? It would be best to include the denominator to determine the response rate as if it is only 1-2% of all subjects, there exists significant response bias.
2) Table 2 the n=109. However, the hand hygiene policy states there were 118. All of the percentages do not work with an n=109. Please revise and correct.
3) Throughout the tables, it is unclear what the sample size is for each variable. While 97% "Increasing basin accessibility" sounds great for institution utilization (Table 4), is this out of 25 responses or all 364 survey responses? Surely there are some surveys that were not fully filled out for all questions which would change the sample sizes.
4) All the tables need to define what the data is showing. For instance, I believe Table 2 reports values as "n (%)". If this is correct, it should be stated as such in the table. What does Table 5 report? Is this a Likert scale out of 5? If so, are these values reported as "Median (SD)"? Please correct.
